# Synthetic Non-Coding RNA for Suppressing mTOR Translation to Prevent Renal Fibrosis Related to Autophagy in UUO Mouse Model

**DOI:** 10.3390/ijms231911365

**Published:** 2022-09-26

**Authors:** Young-Ah Kim, Hyemin Gu, Mi-Gyeong Gwon, Hyun-Jin An, Seongjae Bae, Jaechan Leem, Hyun Jin Jung, Kwan-Kyu Park, Sun-Jae Lee

**Affiliations:** 1Department of Pathology, School of Medicine, Daegu Catholic University, Daegu 42472, Korea; 2Department of Immunology, School of Medicine, Daegu Catholic University, Daegu 42472, Korea; 3Department of Urology, School of Medicine, Daegu Catholic University, Daegu 42472, Korea

**Keywords:** mTOR, UUO, kidney fibrosis, oligodeoxynucleotides, non-coding RNAs

## Abstract

The global burden of chronic kidney disease is increasing, and the majority of these diseases are progressive. Special site-targeted drugs are emerging as alternatives to traditional drugs. Oligonucleotides (ODNs) have been proposed as effective therapeutic tools in specific molecular target therapies for several diseases. We designed ring-type non-coding RNAs (ncRNAs), also called mTOR ODNs to suppress mammalian target rapamycin (mTOR) translation. mTOR signaling is associated with excessive cell proliferation and fibrogenesis. In this study, we examined the effects of mTOR suppression on chronic renal injury. To explore the regulation of fibrosis and inflammation in unilateral ureteral obstruction (UUO)-induced injury, we injected synthesized ODNs via the tail vein of mice. The expression of inflammatory-related markers (interleukin-1β, tumor necrosis factor-α), and that of fibrosis (α-smooth muscle actin, fibronectin), was decreased by synthetic ODNs. Additionally, ODN administration inhibited the expression of autophagy-related markers, microtubule-associated protein light chain 3, Beclin1, and autophagy-related gene 5-12. We confirmed that ring-type ODNs inhibited fibrosis, inflammation, and autophagy in a UUO mouse model. These results suggest that mTOR may be involved in the regulation of autophagy and fibrosis and that regulating mTOR signaling may be a therapeutic strategy against chronic renal injury.

## 1. Introduction

The prevalence of chronic kidney disease (CKD) is increasing worldwide, and fibrosis is responsible for chronic progressive kidney failure [1]. Previous studies have reported that about 13% of the world’s population suffers from CKD [1]. Fibrosis in the kidney is a pathophysiological process characterized by tubular-interstitial fibrosis (TIF) and glomerulosclerosis, which is the final common pathway for all progressive forms of CKD, ultimately leading to end-stage renal disease [2,3]. Fibrosis is defined as the abnormal accumulation of the extracellular matrix (ECM); it is a pathological feature of chronic diseases and is often closely related to organ dysfunction and organ failure [4,5]. Many events can occur at the cellular and molecular levels in renal fibrosis, such as the activation of interstitial myofibroblasts, epithelial-mesenchymal transition, ECM deposition, microvascular dysfunction, and autophagy [6,7]. Renal TIF is initiated and sustained by many pro-sclerotic factors, including transforming growth factor-β1 (TGF-β1), which can increase the expression of matrix proteins and induce epithelial-mesenchymal transition in the kidney: collagen II and collagen IV [8,9]. Other mechanisms, including oxidative stress, inflammation, mitochondrial damage, endoplasmic reticulum stress, and autophagy, are also involved in the progression of CKD [10]. The mammalian target of the rapamycin (mTOR) signaling pathway has been established to be involved in cellular growth, metabolism, fibrogenesis, regulation of inflammation, and autophagy [11,12,13]. Previous research groups have shown that mTOR signaling stimulates excessive cell proliferation and the enhanced production of TGF-β1, collagen IV, and fibronectin [14,15]. mTOR expression is affected by the signal transducer and activator of transcription 3 (STAT3)-mediated regulation of autophagy [16]. Of the signals inducing autophagy, STAT3 is an important intracellular signaling pathway that may regulate inflammations and proliferation via various cytokines and stimuli [17].

Among several strategies of treatment for renal tubular injury, the oligonucleotide (ODN) technique uses synthetic double-stranded non-coding ribonucleic acids (ncRNAs), also called ODNs, that contain antisense sequences for mTOR messenger ribonucleic acid (mRNA) to block mTOR activity. ODNs have been proposed as effective novel therapeutic tools in specific molecular target medicine for several diseases [18,19,20]. According to several previous studies, ring-type ODNs have a significant effect on regulating protein or transcription factors [21,22,23]. In particular, Kim et al. [22] demonstrated that synthetic oligodeoxynucleotide targeting for TGF-β1 and Smad had anti-fibrotic effects on liver cirrhosis in an animal model. Kim et al. [21] reported that STAT3 decoy ODNs inhibited autophagy associated with kidney TIF and inflammation in mice through the inhibition of anti-fibrotic and inflammatory effects. Additionally, inhibiting mTOR and STAT3 by synthetic ODNs has an anti-fibrotic effect in unilateral ureteral obstruction (UUO)-induced renal injury [24]. Although these ODNs have proven beneficial in several disorders, it has not been demonstrated that blocking mTOR function could attenuate the development of the molecular mechanisms of fibrogenesis, inflammation, and autophagy. Additionally, it remains unclear whether autophagy has a therapeutic effect on renal injury. Thus, it is necessary to evaluate the underlying mechanism of renal injury in a mouse model. This study investigates the role of autophagy, inflammation, and TIF in the kidney, by inhibiting mTOR functions. The ncRNAs were designed to inhibit mRNA expression of mTOR in an UUO-induced kidney injury model.

## 2. Results

### 2.1. Construction of mTOR ODNs

Synthetic ncRNAs for suppressing mTOR translation, also called mTOR ODNs, were designed to target mTOR mRNA. The target sites for mTOR ODNs were selected using the S-Fold program. The mTOR ODNs were designed as double-stranded ring-type structures to stabilize the structure from the nuclease (Figure 1). The mTOR ODNs contain a complementary sequence of mTOR mRNA (5′-GAGUUCACACACGUCAAGGAC-3′).

### 2.2. mTOR ODNs Attenuate UUO-Induced Kidney Tubular Injury, Tubulointerstitial Inflammation, and Fibrosis

In this study, the effect of mTOR ODNs on the morphological change caused by UUO surgery was investigated via hematoxylin and eosin (H&E) and Masson’s trichrome staining (Figure 2A,B). There were no significant morphological changes in the normal control (NC) and mTOR groups (Figure 2A). In the UUO and scrambled (Scr) groups, interstitial injury, including tubular distension, atrophy, and TIF with inflammatory cell infiltration, was observed. Compared to the UUO group, mTOR ODNs markedly reduced these morphological changes in the UUO + mTOR group. Masson’s trichrome staining was performed to evaluate collagen deposition. The collagen fiber was deposited in the renal tubules, and renal TIF could be detected in the UUO group, whereas the expression of collagen was clearly decreased in the UUO + mTOR group (Figure 2B). 

To investigate the effects of mTOR ODNs in UUO-induced tubular injury, the expression of neutrophil gelatinase-associated lipocalin (NGAL) and kidney injury molecule-1 (Kim-1) was observed by immunohistochemical (IHC) staining (Figure 2C,D). NGAL and Kim-1 are biomarkers of tubular injury in the kidney [25]. The injured renal tubules showed significantly increased NGAL and Kim-1 expression in the UUO kidney, which could be suppressed by mTOR ODN treatment (Figure 2C,D). The expression of both markers was increased in the UUO and UUO + Scr groups, whereas the UUO + mTOR ODN treatment effectively inhibited NGAL and Kim-1 expression (Figure 2C,D). 

In addition, Western blot analysis was performed to confirm the regulation effects of mTOR ODNs on TIF and inflammation in UUO mice (Figure 3). The UUO model is a well-known progressive renal TIF model [26]. The results of the Western blot analysis showed increased expressions of inflammatory-related transcription factor (p-STAT3), inflammatory cytokines (IL-1β, TNF-α), NGAL, and fibrosis-related markers (α-SMA, fibronectin) in UUO-induced renal fibrosis (Figure 3). In contrast, the inhibition of mTOR by ODNs suppressed the expression of p-STAT3, IL-1β, TNF-α, NGAL, α-SMA, and fibronectin (Figure 3). Taken together, these data indicate that tubular injury, inflammation, and TIF can be down-regulated by mTOR ODN treatment in a UUO mouse model. 

### 2.3. Regulation Effects of Autophagy by mTOR ODNs

This study performed immunofluorescence staining to confirm whether autophagy was promoted in UUO-induced mTOR signaling pathway activation (Figure 4). Phospho-S6 kinase (p-S6K) is a protein that is closely related to the mTOR signaling pathway. mTOR activation can be determined by the increased phosphorylation of S6K [27]. In this study, the expressions of autophagy-related genes (Atg) 5-12 and p-S6K were significantly increased in UUO-induced renal fibrosis. However, the expression of both markers decreased in the UUO model injected with mTOR ODNs (Figure 4A). Additionally, Western blot results show that the administration of Scr ODN increased Beclin1 and light chain 3 (LC3) expression, whereas mTOR ODN injection downregulated Beclin1 and LC3 in the UUO kidney (Figure 4B). Taken together, these data suggest that the suppression of mTOR by ODN can modulate autophagy in UUO-induced renal fibrosis. 

### 2.4. mTOR ODNs Regulated the mTOR Signaling Pathway through p70S6 Kinase and 4E-BP1

To identify the molecular mechanisms of mTOR ODNs, the expression of mTOR-related signaling molecules, including P70, phospho (p)-4E-BP1, was investigated through Western blot analysis and p-p70S6Kinase (K) by immunofluorescence staining (Figure 5). The expression levels of p-p70S6K, p70, and p-4E-BP1 were significantly increased in the UUO and UUO + Scr groups compared with the NC and mTOR groups. However, synthetic mTOR ODN inhibited UUO-induced p-p70S6K, p70, and p-4E-BP1 expression. In addition, there was no significant difference between the groups in mTOR expression levels.

## 3. Discussion

Among kidney diseases, CKD is a chronic progressive disease characterized by abnormalities of kidney structure or function that is present for three months and has implications for health [1]. As a significant portion of the kidney is made up of the renal tubules and interstitium, they are the major sites affected by injuries [28]. Renal fibrosis and inflammation are common pathological results of CKD. Tissue fibrosis is closely associated with chronic inflammation in many pathologies, resulting in functional damage and ultimately leading to terminal renal failure [29]. The progressive property of CKD is associated with the constant loss of renal tissue and its replacement by ECM, culminating in organ fibrosis and failure [30]. The UUO model is a representative animal model of obstructive nephropathy, and it is characterized by progressive tubular interstitial fibrosis [31]. Fibrosis is a pathological feature of CKD and is related to kidney dysfunction due to the abnormal accumulation of ECM [5]. Matrix proteins consist of interstitial collagens and fibronectin [5]. In addition, myofibroblasts express α-smooth muscle actin [12]. Currently, research on the treatment of chronic renal disease is being actively conducted. Therefore, this study attempted to determine whether mTOR ODN could inhibit the expression of α-SMA and fibronectin in UUO-induced renal injury by Western blot analysis. 

Among the many regulatory factors involved in CKD, the mTOR signaling pathway plays an important role in the regulation of inflammatory responses, myofibroblast activation, and deposition during inflammatory conditions [12,32]. Rapamycin, which is an mTOR inhibitor, has shown that it can attenuate inflammation and renal fibrosis in various types of kidney disease [33]. Additionally, mTOR can modulate the activity of inflammatory transcription factors, including STAT3 [34]. IL-1β and TNF-α are potent inflammatory mediators with endocrine effects in chronic inflammation [35]. The activated STAT3 transcription factor is also known to induce inflammatory cytokines, such as TNF-α and IL-1β [36]. As shown in Figure 3, the NC group showed little expression of p-STAT3, IL-1β, and TNF-α in the Western blot analysis. In addition, there was an increased p-STAT3-mediated up-regulated expression of inflammatory cytokines, IL-1β, and TNF-α in the UUO and UUO + Scr groups. However, the expression level of inflammatory cytokines was inhibited by ODN injection.

mTOR signaling and adenosine-monophosphate activated-protein kinase (AMPK) are major regulators of autophagy in acute kidney injury (AKI) induced by cisplatin treatment [37]. In that case, mTOR is a negative regulator of autophagy in the case of AKI [37]. However, there are also opposing views on the role of mTOR in the regulation of autophagy [38,39]. Autophagy has long been considered a target site as a modulatable cellular process in age-related diseases [40]. As mentioned previously, the effects of synthetic ODN targeting for STAT3 and targeting for mTOR and STAT3 were explored [21,24]. Therefore, we tried to inhibit the mTOR function by designing mTOR ODNs. 

In order to reach a therapeutic target and maintain the therapeutic effect, it is important to obtain nuclease resistance for ODNs. To overcome the digestion of ODNs by nuclease activity, some modifications have been developed [41]. Based on previous studies of ODNs, the current study designed a ring-type and double-stranded-type structure to prevent the restriction of ODNs by nuclease digestion in vivo [42]. Previous studies have shown that ring-type ODNs were more stable and effective in nuclease degradation than linear-type ODNs [43,44,45]. Furthermore, Kunugiza et al. [46] demonstrated that the stability and binding ability of the ribbon-shaped ODNs were superior to that of the linear structure. 

Intracellular delivery can be a major barrier of ODNs to effective activity in target cells [47] when internalized ODNs are taken up by endocytosis and traffic to the nucleus [47]. Furthermore, the cell-surface receptors, including scavenger receptors [48], Toll-like receptors [49,50,51], and integrins [52], have been suggested to bind ODNs. In previous studies, we confirmed the stability of synthetic ODNs by in vitro and animal models [22,23]. The cells treated with FITC-labeled TGF-β1/Smad ODNs appeared strongly fluorescent in both the cytoplasm and nucleus of cells after 48 h of transfection [22]. Moreover, administered intravenously FITC-labeled ODNs were shown in the cytoplasm and nucleus of liver cells [23].

This study investigated the effects of mTOR ODNs as inhibiting the mTOR signaling pathway by Western blot analysis and immunofluorescence staining. The Western blot and immunofluorescence staining results demonstrated that mTOR ODN successfully inhibited p70, p-STAT3, p-4E-BP1, p-P70S6K, and p-S6K, which are closely related to mTOR signaling pathways in an animal model of UUO.

The progressive property of CKD is associated with the constant loss of renal tubules and the replacement of collagen tissue with fibrosis [30]. In this study, Western blot analysis was performed to confirm the expression of α-SMA and fibronectin in UUO-induced renal fibrosis (Figure 3). Masson’s trichrome stain was conducted to evaluate collagen deposition in the kidney (Figure 2). As shown in Figure 2, increased collagen deposition in the tubulointerstitial area was noted in the UUO group and decreased in the mTOR ODNs treatment group. Additionally, the results of the Western blot analysis showed that UUO surgery induced α-SMA and fibronectin expression, and they were reduced by mTOR ODN treatment. According to Woodcock et al. [12], as the mTOR/4E-BP1 axis represents a critical signaling node during fibrogenesis, this signaling axis can be a target for anti-fibrotic treatment. Furthermore, Shigematsu et al. [53] and Nishioka et al. [54] have shown that mTOR inhibitors attenuate fibrosis. mTOR signaling inhibition via the mTOR inhibitor, everolimus, had anti-fibrotic effects in the kidney.

IHC staining was performed to evaluate renal tubular injury in UUO-induced renal damage via the expression levels of Kim-1 and NGAL (Figure 2C,D). Additionally, overexpression of NGAL in the UUO mouse model and a significant decrease in the mTOR ODN group were confirmed through Western blot analyses (Figure 3). Inflammation is important in CKD and plays an important role in renal function. This study thought that mTOR ODN would effectively suppress various inflammatory cytokines. To prove this hypothesis, Western blot analysis was used to evaluate the expression of IL-1β and TNF-α (Figure 3). IL-1β and TNF-α are typical inflammatory cytokines. 

Autophagy has long been targeted as a modulatable cellular process with vast consequences in age-related diseases, including cancer and neurodegeneration, as well as in lifespan extension [40]. There are two distinct mTORC1/2 complexes, both of which regulate autophagy [40]. The role of autophagy in renal fibrosis remains unclear. Opinions on whether autophagy protects or damages the kidney are controversial. The interaction between autophagy and renal inflammation is not completely understood. Several study groups have suggested that autophagy serves a dual purpose. In a previous study, TGF-β1 induced autophagy in primary mouse renal tubular epithelial cells and human renal proximal tubular epithelial cells, and autophagy also regulated TGF-β expression and suppressed renal fibrosis induced by UUO [55]. Some studies have reported that when autophagy is activated persistently, renal cell death pathways can be induced, which may lead to kidney damage [56,57]. However, other study groups have suggested that autophagy in the kidney is vital for homeostasis and that downregulation of autophagy is associated with protective effects on renal tubular cells [58,59,60]. When autophagy was suppressed in a UUO mouse, fibrosis and inflammatory activity decreased [21,58]. In this study, as an autophagy marker, Atg 5-12 Beclin1, and LC3 were evaluated via immunofluorescent stains and Western blot analysis (Figure 4). The autophagy markers were activated in the UUO and UUO + Scr groups and alleviated in the mTOR ODN-injected group. The dual function of autophagy may be associated with the duration of renal injury and the stage of CKD. Autophagic downregulation can be observed in the early stages of diabetes and is also often observed in renal fibrosis in the late stages of diabetic nephropathy [61]. As a novel therapeutic approach, synthetic mTOR ODNs have been applied to suppress the function of mTOR. mTOR ODNs were designed as complementary nucleic acid fragments that are specifically triggered through ribonuclease H cleavage of the target mRNA in the nucleus [62]. 

In summary, this study demonstrated the role of mTOR ODNs in UUO-induced renal damage, especially inflammation and fibrosis. It may provide a theoretical basis for anti-fibrotic and anti-inflammatory treatment in clinical practice. This study suggested that the inhibition of the mTOR signaling pathway through ODNs can play a protective role in TIF in the damaged kidney. Therefore, the mTOR signaling pathway can be a new therapeutic target for preventing the progression of renal fibrosis and inflammation.

## 4. Materials and Methods

### 4.1. Synthesis of Ring-Type ODNs

The target sites of mTOR mRNA for ODNs were selected via the sequential overlap simulation of secondary structures using the S Fold program web server at http://sfold.wadsworth.org, accessed on 11 February 2020 [63]. A ring-type with a double-stranded shape was synthesized to stabilize the structure from the nuclease (Figure 1). ODNs were synthesized by Macrogen Co. Ltd. (Seoul, Korea). The sequences of mTOR mRNA and the Scr ODNs used in this work are listed in Table 1 (target sites of consensus binding sequences are underlined). Scr ODNs and mTOR ODNs were annealed for six hours, while the temperature decreased from 80 °C to 25 °C. Following the addition of T4 ligase (1U, Takara Inc., Kusatsu, Japan), the ODNs were incubated for 16 h at 16 °C to obtain covalent ligation for ring-type ODN molecules.

### 4.2. Animal Model and Transfect ODNs

Six-week-old male C57BL/6 mice were purchased from Samtako (Osan, Korea) and housed individually in cages within an environment of controlled humidity (55%) and temperature (22 ± 2 °C) under a 12 h/12 h light-dark cycle. Then, one week after acclimatization, 35 mice were randomly divided into five groups, with seven mice per group. The first group was the NC group. The second group was injected with synthetic mTOR ODNs (the mTOR group). The third group underwent UUO surgery (UUO group). The fourth group contained mice that underwent UUO surgery and were injected with Scr ODNs (the Scr group). The fifth group comprised mice that underwent UUO surgery and were injected with synthetic mTOR ODNs (the UUO + mTOR group). After seven days of acclimatization, the UUO operation was performed at day 3 after anesthetizing the mice. The abdominal cavity was exposed by incision, and the left ureter was ligated with 5-0 silk sutures at two sites: the distal and proximal ureter. The mTOR ODNs (10 μg) and Scr ODNs (10 μg) were injected into the tail veins two days before ureteral ligation surgery, and two days and five days after the UUO operation, by an in vivo gene delivery system (Mirus Bio, Madison, WI, USA). Seven days after the UUO operation, the mice were sacrificed (Figure 6). 

### 4.3. Histological Analysis

After harvesting, all the collected kidney tissues were immediately fixed in 10% formalin at room temperature. The fixed kidney tissues were dehydrated with ethanol, removed with xylene, and placed in paraffin. The paraffin-embedded tissues were cut into 4 μm sections for de-paraffinization. The kidney tissue sections were stained with H&E, as well as Masson’s trichrome, based on standard protocols. All slides were examined with a Pannoramic MIDI slide scanner (3DHISTECH Ltd., Budapest, Hungary).

### 4.4. Immunohistochemical Staining

Xylene was used for the deparaffinization of the paraffin-embedded tissue sections. They were dehydrated in progressively reducing concentrations of ethanol, and then treated with 3% hydrogen peroxidase in methanol for 10 min to prevent endogenous peroxidase activity. The kidney tissue sections were placed in 10 mM sodium citrate buffer (pH = 6.0) for 5 min at 95 °C. The last step was repeated using a new 10 mM sodium citrate solution (pH = 6.0). The sections were allowed to remain in the same solution while cooling for 20 min and were then rinsed in phosphate-buffered saline (PBS). The tissue sections were incubated with a primary antibody such as NGAL (1:500 dilution; Santa Cruz Biotechnology, Dallas, TX, USA) or anti-kidney injury molecule-1 (Kim-1, also called Tim-1; 1:3000 dilution; Abcam, Cambridge, UK) for 1 h at 37 °C. 

The signal was visualized using the EnVision System (DAKO, Agilent, SantaClara, CA, USA) for 30 min at 37 °C. As the coloring reagent, 3,30-diaminobenzidine tetrahydrochloride was used, and hematoxylin was used as the counterstain. All slides were examined with the Pannoramic MIDI slide scanner and analyzed with iSolution DT software. The expression levels of the protein were analyzed via the quantification of the captured images, which were examined via a slide scanner with iSolution DT software.

### 4.5. Immunofluorescent Staining and Confocal Microscopy

The kidney tissue sections were placed in a blocking serum (5% bovine serum albumin in PBS) for 30 min at room temperature. The tissue sections were incubated anti-Atg 5-12 (1:200 dilution; Cell Signaling Technology, Inc., Beverly, MA, USA), phospho-S6 Kinase (1:200 dilution; Cell Signaling Technology), and phospho-p70S6 Kinase (1:200 dilution; Cell Signaling Technology) for 2 h at room temperature. After washing with PBS, the slides were incubated with secondary antibodies (1:200 dilution), conjugated with Alexa Fluor 488 and Alexa Flour 555 (Invitrogen, Waltham, MA, USA). The tissue sections were stained with the nucleic acid stain Hoechst 33342 (DAPI, 1:1000) at room temperature for 2 min. The slides were mounted using a mounting medium (DAKO, Agilent, SantaClara, CA, USA). The stained slides were viewed under a NIKON A1+ confocal microscope (Nikon, Tokyo, Japan). After capturing the image, the quantitative analysis of the expression of Atg 5-12 and the p-S6K area of each group was conducted with the iSolution DT software.

### 4.6. Western Blot Analysis

Kidney protein samples were extracted with a lysis buffer (CelLytic™ MT, Sigma-Aldrich, St. Louis, MI, USA). The protein samples were centrifuged at 13,000 rpm at 4 °C for 10 min after incubation on ice for 30 min. The supernatant was then collected, and the protein concentration was measured using a Bio-Rad Bradford kit (Bio-Rad Laboratories, Hercules, CA, USA) at 595 nm using a spectrophotometer. The samples were boiled for 10 min, and equal volumes were loaded on precast gradient polyacrylamide gels (Bolt™ 4–12% Bis-Tris Plus Gels; Thermo Fisher Scientific, Waltham, MA, USA) before being transferred to a nitrocellulose membrane (GenDEPOT, Barker, TX, USA). The membrane was blocked for 1 h at room temperature in 5% bovine serum albumin and incubated overnight with primary antibodies (1:1000 dilution) at 4 °C. Horseradish peroxidase-conjugated secondary antibodies (1:1000 dilution) were used for 1 h at room temperature. Following a repeat of the wash step, the membranes were kept in detection reagents (Thermo Fisher Scientific). The signal intensity was detected using an image analyzer (ChemiDoc™XRS+, Bio-Rad Laboratories) and quantified with Image Lab software (Bio-Rad Laboratories). The protein expressions were normalized to those of glyceraldehyde 3-phosphate dehydrogenase (GAPDH) or α-tubulin expression values. The primary antibodies used in this study were anti-phospho-mTOR (Ser2448), anti-mTOR, anti-p70, anti-phospho-4E-BP1 (Thr37/46), anti-4E-BP1, anti-phospho-STAT3 (Tyr 705), anti-STAT3, anti-Beclin1, anti-LC3B, anti-GAPDH, anti-α-tubulin (Cell Signaling Technology), anti-fibronectin, anti-TNF-α (Abcam), anti-IL-1β, anti-NGAL (Santa Cruz Biotechnology Inc., Dallas, TX, USA), and anti-α-SMA (Sigma-Aldrich).

### 4.7. Statistical Analysis

The statistical analysis of the data complies with the recommendations on experimental design and analysis in pharmacology [64]. All data were presented as mean ± standard error. Statistical significance was determined via one-way analysis of variance with multiple comparison tests using GraphPad Prism 5.0 (GraphPad Software, Inc., San Diego, CA, USA). Tukey’s tests were run only when F achieved *p* < 0.05, and there was no significant variance in homogeneity. Differences with *p* < 0.05 were considered significant.

## Figures and Tables

**Figure 1 ijms-23-11365-f001:**
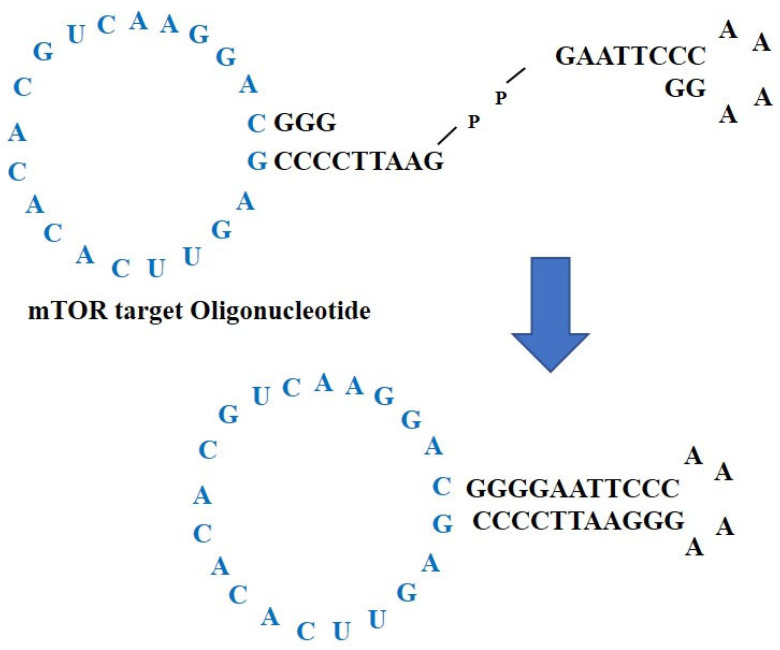
Construction of non-coding RNAs to suppress mTOR translation, mTOR ODNs. The mTOR ODNs have ring-type and double-stranded structures. The ODNs were synthesized by ligating two pieces of ODNs, one of which is mTOR ODNs. The lesion labeled blue is a complementary sequence of mTOR mRNA. ODNs, oligonucleotides; mTOR, mammalian target of rapamycin.

**Figure 2 ijms-23-11365-f002:**
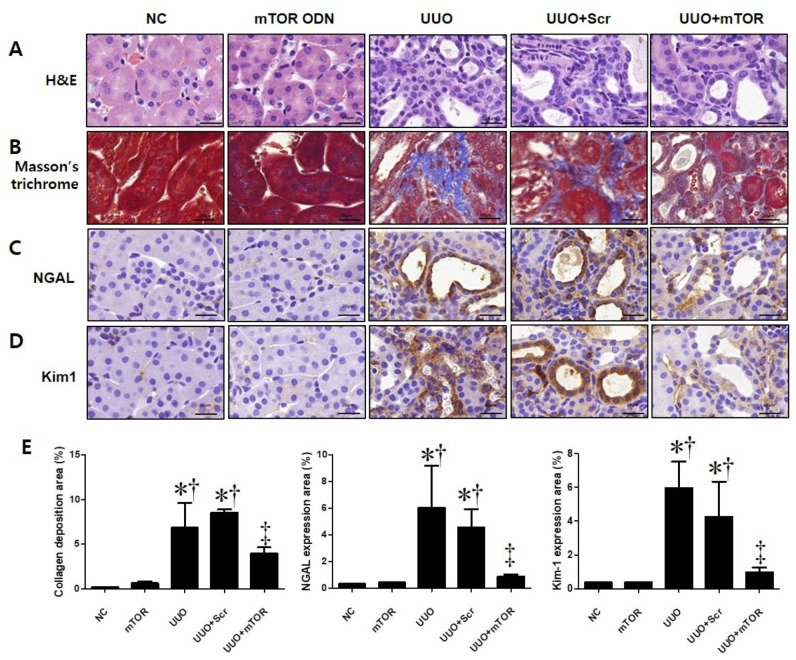
Effects of synthetic mTOR ODNs on UUO-induced damage and tubular injury in mouse kidney tissue. The effects of mTOR ODNs on morphological alterations and tubular injury in UUO kidneys were investigated. Representative images of each group were evaluated for histological changes through (**A**) H&E staining and (**B**) Masson’s trichrome staining. Images of immunohistochemical staining of (**C**) neutrophil gelatinase-associated lipocalin (NGAL) and (**D**) kidney injury molecule-1 (Kim-1). These images are representative of each study group. Scale bar = 20 μm. (**E**) Quantitative analysis of the interstitial collagen area in the trichrome stain, NGAL, and Kim-1 expression areas of each group (*n* = 3) performed at 400× magnification. The graphs show the percentage of collagen deposition, NGAL, and Kim-1 positive areas. The results are expressed as the mean ± SE of three independent determinations. * *p* < 0.05 vs. NC group. ^†^ *p* < 0.05 vs. mTOR group. ^‡^ *p* < 0.05 vs. UUO or Scr group. UUO, unilateral ureteral obstruction; ODNs, oligonucleotides; mTOR, mammalian target of rapamycin; NC, normal control; mTOR, injected with mTOR ODNs; UUO, UUO surgery; UUO + Scr, underwent UUO surgery and injected with scrambled ODNs; UUO + mTOR, underwent UUO surgery and injected with mTOR ODNs.

**Figure 3 ijms-23-11365-f003:**
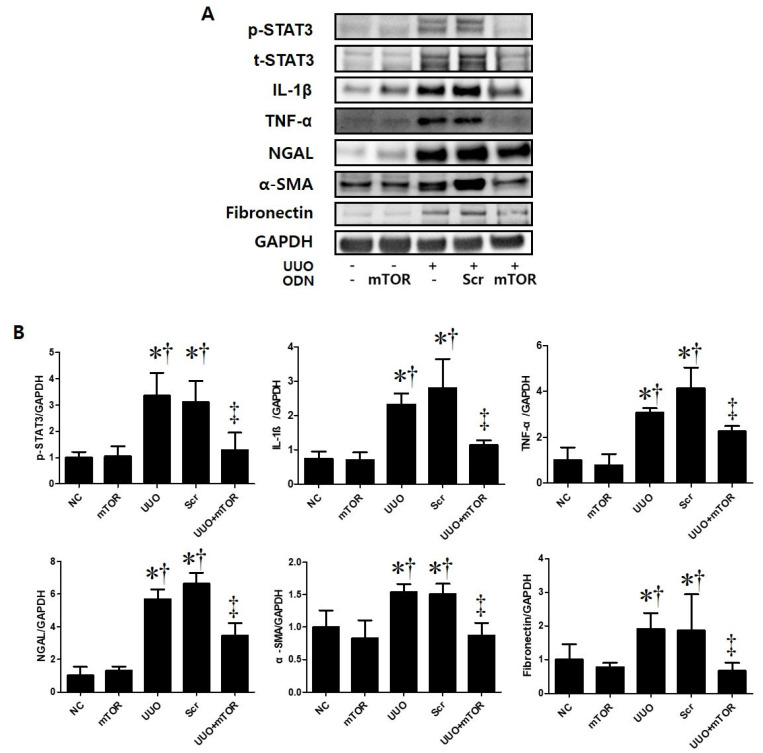
The mTOR ODN significantly attenuated UUO-induced inflammation, tubular injury, and fibrosis. (**A**) The results of Western blot analysis showed that mTOR ODN attenuated the expression of the phosphorylated signal transducer and activator of transcription 3 (p-STAT3), IL-1β, TNF-α, NGAL, α-smooth muscle actin (α-SMA), and fibronectin in UUO mouse model of kidney tissue. (**B**) The expression levels of p-STAT3, IL-1β, TNF-α, NGAL, α-SMA, and fibronectin were quantified. Representative images from each group. The results are expressed as the mean ± SE of three independent determinations. * *p* < 0.05 vs. NC group. ^†^ *p* < 0.05 vs. mTOR group. ^‡^ *p* < 0.05 vs. UUO or Scr group. UUO, unilateral ureteral obstruction; ODNs, oligonucleotides; mTOR, mammalian target of rapamycin; NC, normal control; mTOR, injected with mTOR ODNs; UUO, UUO surgery; UUO + Scr, underwent UUO surgery and injected with scrambled ODNs; UUO + mTOR, underwent UUO surgery and injected with mTOR ODNs.

**Figure 4 ijms-23-11365-f004:**
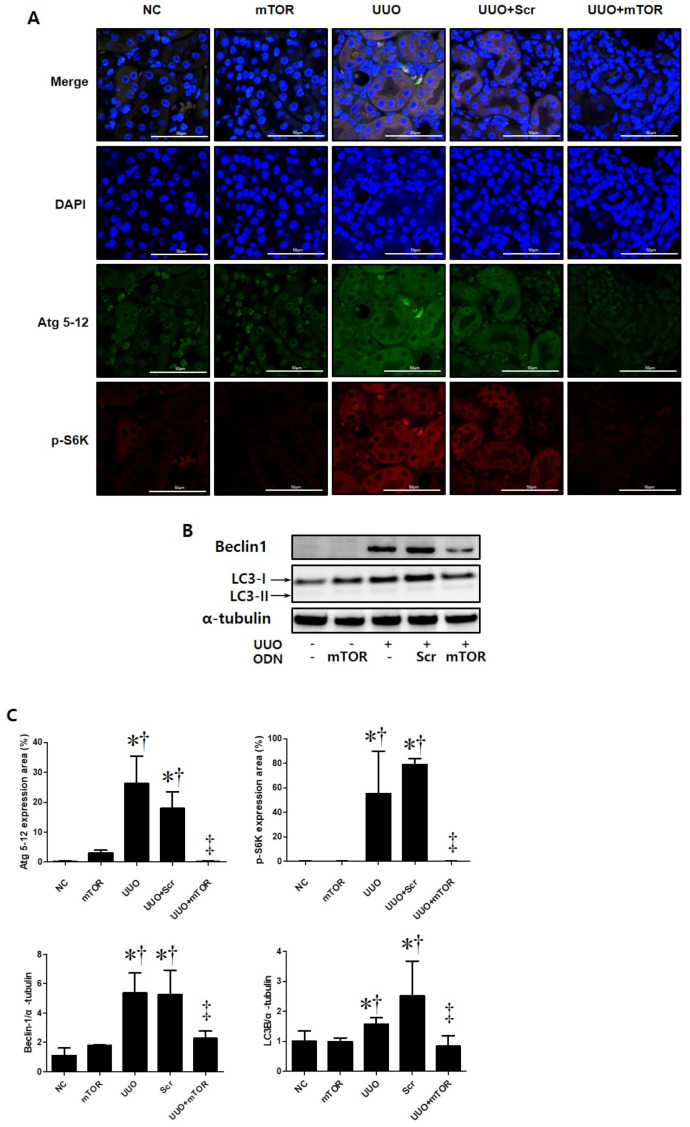
The mTOR ODN inhibited the expression of markers that are important for autophagy and mTOR signaling. (**A**) Representative immunofluorescence double-stain images showed that treatment with the mTOR ODN suppressed autophagy-related gene (Atg) 5-12 (Alexa Fluor 488, green) and phosphor-S6 kinase (p-S6K) (Alexa Fluor 555, red) expression in UUO-induced kidney damage. Nuclei were labeled with Hoechst 33342, DAPI (blue). Scale bar = 50 μm. (**B**) Western blot analysis results showed that ODNs inhibited the expression of Beclin1 and LC3B, which are related to autophagy. (**C**) Quantitative analysis of the expression of Atg 5-12 and the p-S6K area of each group (*n* = 3), Beclin1, and LC3B. The results are expressed as mean ± SE of three independent determinations. * *p* < 0.05 vs. NC group. ^†^ *p* < 0.05 vs. mTOR group. ^‡^ *p* < 0.05 vs. UUO or Scr group. UUO, unilateral ureteral obstruction; ODNs, oligonucleotides; mTOR, mammalian target of rapamycin; NC, normal control; mTOR, injected with mTOR ODNs; UUO, UUO surgery; UUO + Scr, underwent UUO surgery and injected with scrambled ODNs; UUO + mTOR, underwent UUO surgery and injected with mTOR ODNs; DAPI, 4′,6-diamidino-2-phenylindole.

**Figure 5 ijms-23-11365-f005:**
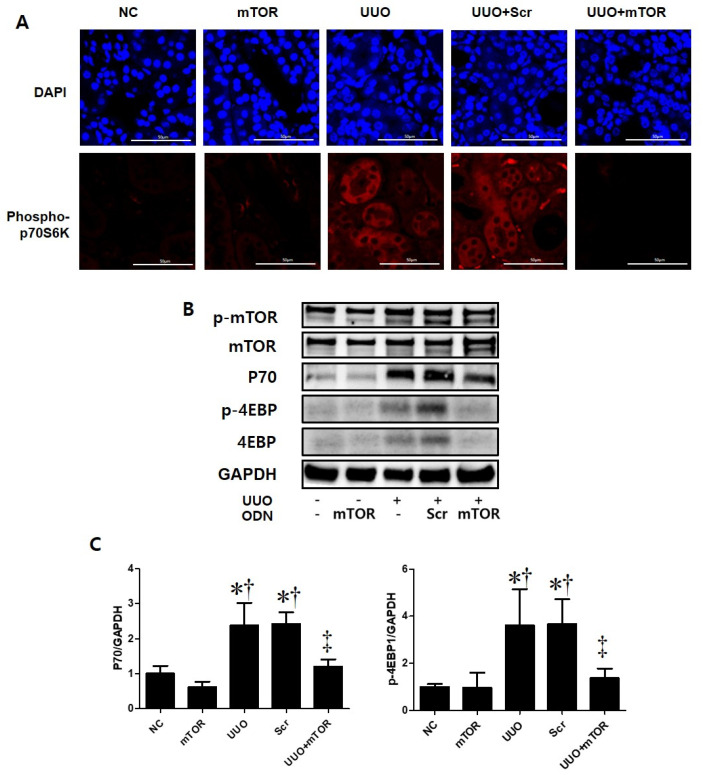
Induction of the mTOR pathway in a UUO mouse model with mTOR ODN injection. (**A**) Representative immunofluorescence images showed the effect of mTOR ODNs on UUO-induced activation of the mTOR signaling pathway. The phospho-p70S6kinase was labeled with Alexa Fluor 555 (red). Nuclei were labeled with DAPI (blue). Scale bar = 50 μm. (**B**) Western blot analysis results showed that synthetic mTOR ODN regulated mTOR signaling pathway-related molecules, p70 and p-4E-BP1. (**C**) The results are expressed as the mean ± SE of three independent determinations. * *p* < 0.05 vs. NC group. ^†^ *p* < 0.05 vs. mTOR group. ^‡^ *p* < 0.05 vs. UUO or Scr group. UUO, unilateral ureteral obstruction; ODNs, oligonucleotides; mTOR, mammalian target of rapamycin; NC, normal control; mTOR, injected with mTOR ODNs; UUO, UUO surgery; UUO + Scr, underwent UUO surgery and injected with scrambled ODNs; UUO + mTOR, underwent UUO surgery and injected with mTOR ODNs; DAPI, 4′,6-diamidino-2-phenylindole.

**Figure 6 ijms-23-11365-f006:**
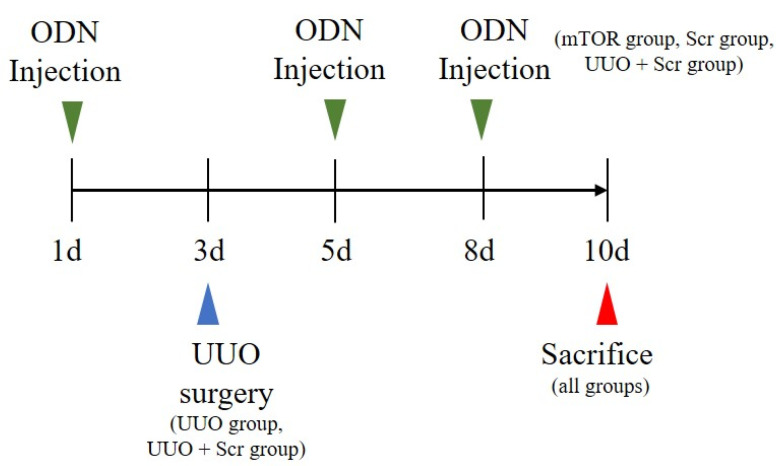
UUO surgery and synthetic mTOR ODN treatment scheme. Except for the mice in the NC and mTOR groups, all mice underwent UUO surgery on day 3. The mTOR group, UUO + mTOR group, and UUO + Scr group received tail vein injections of mTOR ODNs or Scr ODNs on days 1, 5, and 8. NC, normal control; UUO, unilateral ureteral obstruction; ODNs, oligonucleotides; Scr, scrambled; mTOR, mammalian target of rapamycin.

**Table 1 ijms-23-11365-t001:** Sequences of the ODNs used in this study. The target site of the mTOR RNA sequence is underlined.

Decoy	Sequence
mTOR	5′-CCCGAGUUCACACACGUCAAGGACGGGGAATTCCCAAAAGG-3′
Scr	5′-GAATTCAATTCAGGGTACGGCAAAAAATTGCCGTACCCTGAATT-3′

mTOR, mammalian target of rapamycin; Scr, scrambled; ODNs, oligonucleotides.

## Data Availability

Not applicable.

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
