# Peer review of "Synthetic Non-Coding RNA for Suppressing mTOR Translation to Prevent Renal Fibrosis Related to Autophagy in UUO Mouse Model"

_ijms, 2022, doi:10.3390/ijms231911365_

Round 1

Reviewer 1 Report

This group has already published a similar study in Molecules in Feb 2022. Please can the authors elaborate on how this study is different from that manuscript.

Due to the significant overlap between the published study and this manuscript, I do not recommend this article for publication.

Author Response

Please see the attatched file. Thank you.

Reviewer 2 Report

The work by Kim et al designed an antisense oligonucleotide (ASO) against mTOR to treat renal fibrosis in a unilateral ureteral obstruction (UUO) mouse model. The authors showed that the mTOR ASO could attenuate renal fibrosis via inhibition of inflammation and autophagy in mouse. Overall, this manuscript is well written, most of the conclusions are supported by the experimental results. However, this study has some major flaws that prevent its publication at this point:

Major 1: The ASO designed by Kim et al is targeting mTOR and should prevent mTOR translation and reduce mTOR expression. However, in Figure 5B, in all groups (even with mTOR ASO injected groups), mTOR and p-mTOR expression did not show differential expression. This result shows that reduction of inflammation and autophagy by the designed mTOR ASO are probably off-target effects. 

Major 2:  Related to Major 1, to solve off-target effect issue of ASOs, most ASO studies use two or three ASOs that are complementary to different sequences of the target RNA. The authors need to have another mTOR ASO to prove that all observations are really "on-target". Otherwise, the authors shouldn't claim the suppression of renal fibrosis is via inhibition of mTOR translation.

Some minor issues:

Minor 1: Please make group labels in all figures consistent: mTOR or mTOR ODN (in Figure 2, Figure 4 and Figure 5)?  

Minor 2: Figure 4B western blot is lack of UUO and ODN labeling.

Reviewer 3 Report

Authors provide an interesting study, which is a continuation of a previous work (Jung HJ, An HJ, Gwon MG, Gu H, Bae S, Lee SJ, Kim YA, Leem J, Park KK. Anti-Fibrotic Effect of Synthetic Noncoding Oligodeoxynucleotide for Inhibiting mTOR and STAT3 via the Regulation of Autophagy in an Animal Model of Renal Injury. Molecules. 2022 Jan 25;27(3):766. doi: 10.3390/molecules27030766). In this manuscript is it shown that mTOR ODNs reduce UUO-induced kidney damage, based on histological findings, biochemical markers (NGAL, Kim-1), inflammatory molecules (IL-1, TNF) and modulate autophagy.

Minor remarks:

1) there are some minor errors related with spelling (line 23, 28, 287),

2) some sentences are not ended/missing words, i.ex. 'diabetic kidney' DISEASE (line 279),

3) Can you please add more data that mTOR inhibitors were shown to inhibit fibrosis? (i.ex. Shigematsu T, Tajima S, Fu R, Zhang M, Itoyama Y, Tsuchimoto A, Egashira N, Ieiri I. The mTOR inhibitor everolimus attenuates tacrolimus-induced renal interstitial fibrosis in rats. Life Sci. 2022 Jan 1;288:120150. doi: 10.1016/j.lfs.2021.120150

Nishioka S, Ishimura T, Endo T, Yokoyama N, Ogawa S, Fujisawa M. Suppression of Allograft Fibrosis by Regulation of Mammalian Target of Rapamycin-Related Protein Expression in Kidney-Transplanted Recipients Treated with Everolimus and Reduced Tacrolimus. Ann Transplant. 2021 Jan 12;26:e926476. doi: 10.12659/AOT.926476).

Reviewer 4 Report

Authors designed an anti-CKD treatment based in the use of a circular non-coding oligonucleotide against mTOR in a model mouse of unilateral ureteral obstruction. The immunohistological work is conclusive but the molecular biology is sub-standard. Thus, although interesting, the manuscript should be greatly improved to achieve the level required for publishing in IJMS.

Major concerns.

1.- My major concern is that the effects described on the expression of inflammatory/fibrogenis matkers after treatment with anti-mTOR oligo seems to be evident in the UUO groups only. Can authors confirm this finding? Furthermore, they should discuss this point more in deep

2.- Authors should explain more in deep how the anti-mTOR ODN was designed and constructed.  What was the mTOR sequence targeted by the ODN? What are the advantages of using two ligated oligos instead of synthesizing and autoligating a single one? . How is the Scr control folded?  Furthermore, Figure 1 is misleading since it seems that the second oligo is ligated to the long tail of the first one.

3.- Figure 6 is incomplete and lacks all the experimental groups

4.- Authors should discuss more in deep the advantages of circular oligonucleotives vs. the lineal ones.

5.- Authors should describe in deep how the image analysis was done, specially of the IF images (Atg 5-12 and p-S6K in Figure 4?)

Minor concerns.

-Figure 1 (in the text) is refered as Figure 2 in the M&M section (line 295) .

-The S-Fold program should be referenced. Furthermore, the individual subprogram and the settings used should be cited.

-In the M&M section authors state that a confocal microscope was used for the immunohistological work. To my eye, the pictures correspond to “normal” IF captures and authors should confirm this. Furthermore, more details on the microscope used should be given

Round 2

Reviewer 4 Report

Authors have made a great effort to improve the manuscript acording to my requeriments, but I still have a concern over this work, relative to my request for a deeper discussion on the advantages of circular oligos vs. linear ones. Since, in my opinion, this is a critical point that could be of interest to many researchers I would like to see included a more substantial discussion on this topic and not a single reference from the same lab. Ideally the discussion should include "in vitro" (cell culture) and animal models and if possible a comparison with other form of stabilizing oligos (chemical derivatives, end-blockers...).

Author Response

Author's Note is attatched.
